



# High resolution modeling of gaseous methylamines over a polluted region in China: Source-dependent emissions and implications to spatial variations

Jingbo Mao[1], Fangqun Yu[1,2*], Yan Zhang[1*], Jingyu An[3], Lin Wang[1], Jun Zheng[4], Lei Yao[1], Gan Luo[2], Weichun Ma[1], Qi Yu[1], Cheng Huang[3], Li Li[3], and Limin Chen[1]

[1]Shanghai Key Laboratory of Atmospheric Particle Pollution and Prevention, Department of Environmental Science and Engineering, Fudan University, Shanghai 200433, China

[2] Atmospheric Sciences Research Center, State University of New York, 251 Fuller Road, Albany, New York 12203, USA

[3] Shanghai Academy of Environmental Sciences, Shanghai 200233, China

[4] School of Environmental Science and Engineering, Nanjing University of Information Science and Technology, Nanjing 210044, China

* Corresponding authors: F. Yu (fyu@albany.edu) and Y. Zhang (yan_zhang@fudan.edu.cn)

**Abstract**: Amines have received increasing attention in recent years because of their potential role in new particle formation in the atmosphere and their impact on aerosol chemistry. High concentrations of amines are expected to be limited to the vicinity of source regions due to their short lifetime, highlighting the necessity of having a better understanding of contributions of emissions from different source types. This study presents the first high-resolution model simulation of methylamines concentrations on a regional scale over the Yangtze River Delta region in east China. The WRF-Chem with nested grids is used in model simulations. In contrast to the very limited existing modeling studies that assumed a fixed ratio (FR) of amines to total ammonia emission, we derive source-dependent ratios (SDR) that distinguish C1-amine ($CH_3NH_2$), C2-amines ($C_2H_7N$), C3-amines ($C_3H_9N$) emissions from five different source types (agriculture, residential, transportation, chemical industry, and other industry). The amines-to-ammonia mass



emission ratios, estimated from previous measurements, are 0.026, 0.0015, 0.0011, 0.0011, and

0.0011 for C1-amine, 0.007, 0.0018, 0.0015, 0.01, and 0.0009 for C2-amines, and 0.0004, 0.0005,

0.00043, 0.0006, and 0.0004 for C3-amines for chemical-industrial, other industrial, agricultural,

residential, and transportational sources, respectively. The simulated concentrations of C1-, C2-,

and C3-amines, based on both FR and SDR, have been compared with field measurements at a

suburban site in Nanjing and at an urban site in Shanghai, China. SDR substantially improves the

model's ability in capturing the observed concentrations of methylamines. C1-, C2-, and C3-

amines concentrations in the surface layer in the Yangtze River Delta region are generally in the

range of 2-20 pptv, 5-50 pptv, and 0.5-4 pptv. Vertically, the concentrations of C1-, C2-, and C3-

amines decrease quickly with altitude, dropping by a factor of ~10 from the surface to ~900 hPa.

Results from the present study are critical to evaluating potential roles of amines in nucleation and

chemical processes in polluted air.

## 1. Introduction

Gaseous amines may play an important role in new particle formation and growth based on

chamber experiments, theoretical calculations, and field observations (Kurtén et al., 2008;

Almeida et al., 2013; Zhao et al., 2011; Erupe et al., 2011; Chen et al., 2012; Yu et al., 2012;

You et al., 2014; Chen et al., 2014; Jen et al., 2016, Olenius et al., 2017). CLOUD (Cosmics

Leaving OUtdoors Droplets) chamber experiments (Almeida et al., 2013) demonstrate that

dimethylamine (DMA) of above 3 pptv can enhance nucleation rate by more than 1000-fold.

Lehtipalo et al. (2016) reported that the growth rate of sub-3 nm particles at a given $H_2SO_4$

monomer concentration was enhanced by a factor of 10 with addition of > 5 pptv DMA,

compared to a factor of 2-3 enhancement when $NH_3$ of > 100 pptv was added. As ubiquitous

atmospheric organic bases, amines can form ammonium salts by acid-base reactions (Murphy

et al., 2007; Kurtén et al., 2014; Lehtipalo et al., 2016; Tao et al., 2016). In addition to dry and

wet deposition, the concentrations of amines in the air decrease through oxidization reactions

with OH, NOx, and ozone (Carl and Crowley, 1998; Murphy et al., 2007; Nielsen et al., 2012),





and uptake by particles (Qiu et al., 2011; Zhang et al., 2011; Qiu and Zhang, 2013). There are about 150 gaseous amines identified in the atmosphere, but little is known about their thermodynamic and kinetic properties and their importance in the atmosphere (Ge et al., 2011). While measurements of amines in different environments (e.g., rural, urban, marine, and forest)

have been reported (Sellegri et al., 2005; Hanson et al., 2011; VandenBoer et al., 2011; Yu and Lee, 2012; Freshour et al., 2014; You et al., 2014; Zheng et al., 2015; Yao et al., 2016), they are very limited, especially in China. Zheng et al. (2015) measured C1-, C2-, and C3- amines at a suburban site of Nanjing, China during the summer of 2012 and they reported an average total amines value of $7.4\pm4.7$ pptv. Similar measurements of amines were conducted at Fudan

University, an urban site in Shanghai, China during the summer of 2015 and the observed mean concentrations of gaseous C1-C6 amines were $15.7\pm5.9$, $40.0\pm14.3$, $1.1\pm0.6$, $15.4\pm7.9$, $3.4\pm3.7$, and $3.5\pm2.2$ pptv, respectively (Yao et al., 2016). The results in both Nanjing and Shanghai suggest that amines-enhanced particle formation and growth may be important in the Yangtze River Delta, one of the highly polluted regions in China.

It is necessary and important to know the concentrations of key amines and their variations in order to understand the role of amines in particle nucleation and growth. In this regard, numerical models can be useful in simulating the distributions of amines on regional or global scales. To our knowledge, only three modeling studies of amines have been reported in the literature, all on a global scale (Myriokefalitakis et al., 2010; Yu and Luo, 2014; Bergman et

al., 2015). Myriokefalitakis et al. (2010) investigated the potential contribution of amines emitted from oceans to secondary organic formation (SOA) formation, assuming total amine emissions to be one-tenth of the oceanic ammonia emissions. They did not consider amines from continental sources and also did not report any simulated concentrations of gaseous amines over oceans. Yu and Luo (2014) studied the global distributions of the most common

and abundant amines in the air: monomethylamine (MMA), dimethylamine (DMA), and trimethylamine (TMA). They used the ratios of MMA, DMA, and TMA to ammonia fluxes given in Schade and Crutzen (1995), but approximate the spatial distributions and seasonal variations of amine emissions following those of ammonia. Bergman et al. (2015) added one single (unified) alkylamine species that has the physical and chemical properties of TMA into




a global aerosol-climate model, and assumed an amine-to-ammonia ratio of 0.0057 kg (amine (N))/kg (ammonia (N)). Due to the lack of information regarding the emission of amines from different sources, these three previous studies (Myriokefalitakis et al., 2010; Yu and Luo, 2014; Bergman et al., 2015) used fixed amines to ammonia ratios to estimate amines emissions. While such an approximation provides a first order of magnitude estimation of amines emission, it may lead to large uncertainties in the model-predicted concentrations of amines, especially their spatial distributions at regional and urban scales. In fact, Yu and Luo (2014) showed that the predicted amines concentrations based on a global model, with amines to ammonia ratios as reported in the literature, are significantly lower than those observed. One possible reason for the model underprediction is the uncertainty in amines emissions near the sites of measurements.

Amines are emitted into the atmosphere from both natural and anthropogenic sources, including animal husbandry, chemical facilities, industry, carbon sequestration, combustion, fish processing, automobiles, sewage, composting operations, vegetation, soil, biomass burning, and the oceans (Ge et al., 2011). In many situations, amines are co-emitted with ammonia, but the ratios of amines to ammonia from various sources may differ significantly and there may also exist stand-alone sources of amines (Kuhn et al. 2011; Zheng et al., 2015). For example, measurements have indicated that industrial amines emission may be important sources in Nanjing (Zheng et al., 2015). Kuhn et al. (2011) concluded that amines in agricultural regions are mainly released from animal housing and grazing animals, in contrast to ammonia, which is mostly emitted into the atmosphere from agricultural fertilizers. Bergman et al. (2015) also pointed out that the direct calculation of amine emissions based on ammonia can skew the spatial extent of the amine emission and emphasized a clear need for improved estimates of amine emissions from different emission sectors.

Apparently, there is a clear need to better understand emissions of amines from various source types and to improve the model simulations of amines concentrations and their spatial distributions. The main objective of this study is to estimate amines emissions from five different source types (chemical industry, other industry, agriculture, residential, and transportation) and simulate spatial distributions of gaseous amines over the Yangtze River



Delta region in China, using recently available amines measurements and up-to-date (year 2014) emission inventories in the region for various emission sectors with 4 km×4 km horizontal resolution. The observational data used to constrain model simulations includes continuous measurements of amines during a one-month period (summer of 2015) at an urban site in Shanghai, China (Yao et al., 2016) and a one-week period (summer of 2012) at a suburban site in Nanjing, China (Zheng et al., 2015).

## 2. Methods

### 2.1 Emission inventory for anthropogenic sources

Anthropogenic particulate and gas emissions for Asia and China are based on INTEX-B (Zhang et al., 2009) and Multiple-resolution emission inventory for China (MEIC) developed by Tsinghua University (http://www.meicmodel.org), respectively. The emissions for $SO_2$, NOx, CO, VOC, $PM_{10}$, $PM_{2.5}$, and primary black carbon and organic carbon are included in the INTEX-B database with 0.5º×0.5º horizontal resolution, and (with $NH_3$ emissions as well) in the MEIC database with 0.25º×0.25º horizontal resolution.

To improve the emission accuracy and spatial resolution for the Yangtze River Delta region, we employ a refined bottom-up emission inventory (4 km×4 km resolution) for the year 2014 developed by the Shanghai Academy of Environmental Sciences (SAES). The SAES 2014 inventory includes anthropogenic emissions from various chemical, industrial, vehicular, shipping, agricultural, and residential sources. The SAES 2014 inventory is updated from the previous work (Huang et al., 2011; Li et al., 2011), which consists of large point sources, industrial, mobile, residential, and agricultural sources. Point sources in this inventory consist of power plants and large industrial combustion and processing sources. The point sources data are obtained from a national environmental statistical database. Mobile sources consist of on-road vehicle, non-road vehicle, and ship emissions. The vehicle volume data, residential fuel combustion, and the activity data of agriculture sources including the amount of livestock and fertilizer consumption are obtained from the statistical yearbooks of the 41 cities in the Yangtze



River Delta. The detailed information about estimation of ship emissions is given in Fan et al. (2015).

For the temporal variations of emissions, we used profiles derived from local investigation for different emissions sources (Tan et al., 2015). For the spatial distributions of various

emissions, ArcGIS was used to distribute area and stack sources in the emission inventory. Stack sources were allocated into grid cells based on their geographical positions. The height of stack emissions ranges from 20 m to 250 m were based on NOx and $PM_{10}$ emission flux (Tan et al., 2015). Mobile, residential, and agricultural emissions were treated as area sources and distributed into corresponding grid cells.

**2.2 Amines emissions**

It is presently impossible to develop either a global or regional bottom-up emission inventory of amines due to insufficient direct measurements. The fixed amines to ammonia ratio assumed in two previous global studies (Yu and Luo, 2014; Bergman et al., 2015) resulted in higher amines concentrations in agricultural areas than in other areas because agriculture

dominates $NH_3$ emissions. However, very high concentrations of amines in an urban site has been reported (Yao et al., 2016), indicating strong amines sources not associated with agricultural activities. A refined amines emission inventory is apparently needed.

Low-molecular-weight amines are the most common among about 150 amines that have been identified so far. The present study focuses on C1-amine ($CH_3NH_2$), C2-amines ($C_2H_7N$),

and C3-amines ($C_3H_9N$). In contrast to previous modeling studies assuming a fixed ratio (FR) of amines to total ammonia emission, we seek to take into account the dependence of C1-, C2-, and C3-amines-to-ammonia ratios on five different source types (chemical industry, other industry, agriculture, residential, transportation). Agriculture includes livestock, biomass burning, soil, and fertilizer usage. Ammonia is emitted from fertilizer plants by volatilization,

which is similar to ammonia volatilization in soil. Hence, we group fertilizer plants into agricultural sources. The chemical-industrial source type includes emissions from petrochemicals, pharmaceuticals, agrochemicals excluding fertilizer plants, paints, fine



chemicals, and solvent use industries, while other industrial type includes power plants, iron and steel mills, cement, carbon sequestration, food industry (e.g., fish processing), and other industry boilers. Residential source type includes cooking, human excreta, and gas (water) disposal, while transportation includes automobiles and ships.

Zheng et al. (2015) simultaneously measured $NH_3$, C1-, C2-, and C3-amines, NOx and $SO_2$ using an aerodyne HR-ToF-CIMS with high time resolution at Nanjing University of Information Science and Technology (NUIST), a suburban site in Nanjing, China from 26 August to 8 September 2012. The high time resolution HR-ToF-CIMS data resolves individual plumes. Zheng et al. (2015) analyzed this data in detail and identified the possible source types

of plumes based on the differences in the concentrations of $SO_2$ and NOx along with wind directions. Table 1 gives the ratios of C1-, C2-, and C3-amines concentrations to that of ammonia for individual plumes with different origins as identified by the authors. The ratios were derived from simultaneously measured ammonia, C1-, C2-, and C3-amines concentrations as reported by them. Table 1 shows the ratios of C1-, C2-, and C3-amines-to–

ammonia in four source types: other industry, agriculture, transportation, and residential. For the chemical industry, Zheng et al. (2015) reported the presence of relatively high amine concentrations (2.6% of MA, 0.7% of C2-amines, and 0.04% of C3-amines) in the ammonia water solution from a local chemical supplier that has been used as absorbent during flue gas treatment. With the above information, the estimated amines to ammonia emission ratios are

0.026, 0.0015, 0.0011, 0.0011, and 0.0011 for C1-amine, 0.007, 0.0018, 0.0015, 0.01, and 0.0009 for C2-amines, and 0.0004, 0.0005, 0.00043, 0.0006, and 0.0004 for C3-amines for chemical-industrial, other industrial, agricultural, residential, and transportational source types, respectively . We would like to acknowledge that the above estimation of amines emissions from different sources is subject to a large uncertainty, mainly due to very limited

measurements available to constrain the estimation. Nevertheless, the above approach represents the first attempt to derive source-type dependent amines to ammonia ratios, which, as we show below, improves the model's skill in simulating concentrations of amines in polluted regions. In the present study, the temporal and spatial distributions of C1-, C2-, and C3-amines follow those of ammonia for different sources.



### 2.3 Model set up and configurations

We employ WRF-Chem (version 3.7.1), a regional multi-scale meteorology model coupled with online chemistry (Grell et al., 2005). Ammonia is simulated in the standard version of WRF-Chem, but amines are not considered prior to this study. To simulate gaseous amines, we

add three new tracers (C1-amine, C2-amines, and C3-amines) in WRF-Chem. The model configurations include Morrison2-mom microphysics (Morrison, H. et al., 2009), RRTMG longwave and shortwave radiation (Clough et al., 2005), Noah land surface, Grell-3 cumulus (Grell and Freitas, 2014), and YSU PBL scheme (Hong et al., 2006). For gas-phase chemistry, we use CB05 scheme (Yarwood et al., 2005). The surface areas of pre-existing particles,

important for the uptake of amines in the atmosphere, are calculated from particle size distributions predicted by an advanced particle microphysics (APM) model embedded into WRF-Chem (Luo and Yu, 2011). The initial and boundary conditions for meteorology are generated from the National Centers for Environmental Prediction (NCEP) Final (FNL) with horizontal resolution at $1° \times 1°$ and time intervals at six hours. The detailed anthropogenic

emissions are described in Section 2.1, and the biogenic emissions are calculated online using MEGAN (Guenther et al., 2006). After emissions, gaseous amines are removed by dry and wet deposition, gas-phase reaction, and aerosol uptake (Yu and Luo, 2014, Bergman et al., 2015). The treatment of deposition, oxidation, and uptake for amines in WRF-Chem/APM follows the approach as described in Yu and Luo (2014).

The WRF-Chem/APM is used for four nested domains simulations with horizontal resolutions of 81, 27, 9, and 3 km (Figure 1) and vertical resolution of 22 layers (from surface to ~11.8 km) with 8 levels below 1.5 km. Domain 1 covers East-Asia and part of south-east Asia. Nested domains 2, 3, and 4 cover a large part of East-China, the Yangtze River Delta (including Nanjing and Shanghai), and Shanghai with the complex underlying surface,

respectively.

Our simulations focus on two periods during which continuous measurements of amines are available: (1) 26 August to 31 August 2012, and (2) 25 July to 25 August 2015. The model spin-up time is 3 days for each case. For each period, two separate simulations were carried



out: one assumes a fixed ratio (FR) of amines to ammonia emissions used in all previous studies (Myriokefalitakis et al., 2010; Yu and Luo, 2014; Bergman et al., 2015), and the other one employs source dependent ratios (SDR) as described in Section 2.2. Table 2 summarizes the four simulation cases: FR2012, SDR2012, FR2015, and SDR2015. For the two FR cases,

the ratios of amines to ammonia emissions for C1-, C2-, and C3-amines for all source types, estimated from the global emission budgets given in Schade and Crutzen (1995), are 0.0017, 0.0007, and 0.0034, respectively. For the two SDR cases, we also carry out a sensitivity study by halving and doubling the ratios given in Table 1.

## 3. Results

### 3.1 Contribution of methylamines emissions from various source types

Ammonia, C1-, C2-, and C3-amines emission rates based on SDR in the Yangtze River Delta for residential, agricultural, other industrial, chemical-industrial, and transportational sources are summarized in Table 3. Ammonia emission rate in the Yangtze River Delta region is 919.61 Gg N yr$^{-1}$, and total C1-, C2-, and C3-amines emission rates based on SDR (FR) are estimated

as 551.88 (1563.34), 849.11 (643.73), and 117.78 (3126.67) Mg N yr$^{-1}$, respectively. The significant difference in the estimated emission rates of amines in the region can be clearly seen, especially for C1- and C3- amines. Based on SDR, the contributions of agricultural, residential, transportational, other industrial, and chemical-industrial sources to domain-averaged methylamines (C1-amine+ C2-amines+ C3-amines) are 66.04%, 30.81%, 1.61%,

0.81%, and 0.73%, respectively. Agricultural source type is the largest contributor for C1-, C2-, and C3-amines, while residential is another main contributor especially for C2-amines (~46%).

The horizontal distributions of C1-amine, C2-amines and C3-amines from different sources and total emission fluxes are presented in Figs. 2-4. The emission fluxes for C1-amine, C2-amines, and C3-amines, respectively, from five sources are mainly in the range of 0.1-10, 1-

25 100, and 0.05-6 Mg N km$^{-2}$ yr$^{-1}$ from residential sources (Figs. 2a-4a), 0.1-50, 0.5-60, and 0.1-8 Mg N km$^{-2}$ yr$^{-1}$ from agriculture (Figs. 2b-4b), 0.01-1, 0.01-3, and 0.01-0.6 Mg N km$^{-2}$ yr$^{-1}$ from other industry (Figs. 2c-4c), 0.01-20, 0. 01-10, and 0.01-0.03 Mg N km$^{-2}$ yr$^{-1}$ from chemical industry (Figs. 2d-4d), and 0.01-0.8, 0. 01-0.6, and 0.01-0.3 Mg N km$^{-2}$ yr$^{-1}$ from





transportation (Figs. 2e-4e). Total emission flux of C2-amines is in the range of 0.1-100 Mg N km$^{-2}$ yr$^{-1}$ over continents in the Yangtze River Delta and below 0.01 Mg N km$^{-2}$ yr$^{-1}$ over ocean near Yangtze River Delta (Fig. 3f). For C1-amine and C3-amines, the total emission fluxes are 0.1-50 Mg N km$^{-2}$ yr$^{-1}$ and 0.1-6 Mg N km$^{-2}$ yr$^{-1}$ and less than 0.01 Mg N km$^{-2}$ yr$^{-1}$ over oceanic area (See Figs. 2f, 4f). As mentioned earlier, we assumed that the spatial distributions of methylamines from five sources (agriculture, residential, transportation, other industry, and chemical industry) to be the same as those of ammonia. As can be seen from Figs. 2f-4f, the horizontal distributions of total C1-, C2-, and C3-amines emission fluxes are different from that of ammonia (not shown), especially over agricultural areas for C2-amines. To assess the effect of amines emission assumptions, comparisons of simulated C1-, C2-, and C3-amines based on SDR approach in the present study with the FR method used in previous studies (e.g., Yu and Luo, 2014) with those observed at a suburban site (NUIST site, Nanjing, China) and an urban site (Fudan site, Shanghai, China) are given in the next section.

## 3.2 Comparisons of simulations with observations

Figures 5-6 compare wind fields and C1-, C2-, and C3-amines concentrations simulated using FR and SDR with measurements at NUIST site in Nanjing, China (FR2012 and SDR2012, Fig. 5) and Fudan site in Shanghai, China (FR2015 and SDR2015, Fig. 6). Simulated wind direction at the NUIST site (Fig. 5a) is overall consistent with observations, so is wind speed at 10 m (Fig. 5b) except that the model overpredicted for 28 August to midday of 29 August. For the Fudan site, model simulations (Fig.6a) generally reproduce observed wind direction, although there exist large differences during some periods. The simulated wind speeds at 10 m (Fig. 6b) are in agreement with observations, except during the periods of August 7-14 and August 23-25. These deviations may be caused by local underlying surface or other physical parameters in the complex urban environment.

Mean values and normalized mean biases (NMBs) are given in Table 4 to summarize the statistics performance of model calculated C1-, C2-, and C3-amines for different cases. Previous global simulations (Yu and Luo, 2014) show general underprediction of the model



(NMB values of -61.4% for C1-amine, -79.9% for C2-amines, and -60.9% for C3-amines), while this study indicates that amines concentrations based on the model with high spatial resolution can also be overpredicted. Overall, simulations based on SDR are in much better agreement with measurements than those based on FR, especially for C2- and C3-amines.

Replacement of FR with SDR improves NMB for C2-amines from -71.5% to 49.12% at the NUIST site and from -96.13% to -37.43% at the Fudan site, while NMB improves for C3-amine from 359.02% to -41.26% at the NUIST site and from 494.28% to 21.34% at the Fudan site. The different performance of the model in the NUIST and Fudan sites is probably due to, but not limited to, uncertainties in meteorology fields, amines emissions and loss processes,

and model resolutions. For C1-amine, both FR and SDR overpredict the concentrations by a factor of 2-3 at the NUIST site, while underpredict by a factor of 3-4 at the Fudan site. A comparison of simulated C1-, C2-, and C3-amines in domain 4 at resolution of 3 km×3 km (blue lines in Figs. 6c-e) with those in domain 3 (9 km×9 km horizontal resolution) (red lines in Figs. 6c-e) shows that the concentrations in domain 4 are generally higher, especially for

peak values. As can be seen from the NMB values, domain 4 values are in better agreement with observations, highlighting the importance of high resolution modeling in resolving the spatial variations in urban environments. It should be noted that the model-predicted C1- and C2-amines at the Fudan site for the period of August 7-19 are much lower than the observed values (Figs. 6c-d), at least partially due to the large deviation of the simulated wind directions

and speeds during the period (Figs. 6a-b).

As mentioned in Section 2.2, there exist large uncertainties in methylamines emissions because of the very limited observations available. To evaluate the effect of uncertainties in emissions on simulated amines concentrations, we also carry out a sensitivity study for the two SDR cases by halving and doubling the five sources ratios simultaneously, defined as 0.5

SDR2012, 2 SDR2012, 0.5 SDR2015, and 2 SDR2015, respectively. In this sensitivity study, only emission ratios were changed, with other processes including deposition, oxidation, and uptake for amines unchanged. For 0.5 SDR2012 and 2 SDR2012, simulations focus on the period from 26 August to 31 August 2012 (the same as the SDR2012 case), while for 0.5 SDR2015 and 2 SDR2015 the simulated period is from 25 July to 31 July 2015 when the model





reproduced relatively well the wind fields (Figure 6). Comparisons of simulated C1-, C2-, and C3-amines concentrations in SDR2012, 0.5 SDR2012, and 2 SDR2012 at the NUIST site (domain 3) and SDR2015, 0.5 SDR2015, and 2 SDR2015 at the Fudan site (domain 4) with measurements are shown in Figs. 7-8, with corresponding NMB values summarized in Table 5. As expected, simulated concentrations of amines are sensitive to the assumed emission ratios and it is clear that the uncertainties in emission ratios can account for a large fraction of difference in the simulated and observed concentrations. It should be noted that, as a result of variations in human activities and/or operation conditions of facilities associated with amines emissions in the real atmosphere, the amines to ammonia emission ratios from a given source sector may vary with time, which may lead to the spikes in the observed amines concentrations that are missed by the model simulations.

The performance of simulations with different wind directions are of varying quality at the Fudan site. Periods with difference in simulated and observed wind directions within -30° to 30° are selected for further comparisons. Figure 9 shows a close comparison of simulated and observed C2-amines concentrations with wind direction between 175° and 240° (Fig. 9a) and other wind directions (Fig. 9b) at Fudan site. It is clear that simulated C2-amines concentrations with wind direction between 175° and 240° where the air mass was coming from residential areas (see Fig. 3a) are more consistent with measurements ($NMB_{d3\_SDR}$= -26.07%, $NMB_{d4\_SDR}$= 7.03%) than those from other wind directions ($NMB_{d3\_SDR}$= -63.52%, $NMB_{d4\_SDR}$= -41.34%), indicating that the SDR-based residential emissions for C2-amines may be reasonable. It can also be seen from Fig. 9 that FR assumption underpredicts C2-amines by 1–2 orders of magnitude with NMB under -90% for any wind direction, highlighting the necessity to use SDR in polluted urban areas such as Shanghai, China.

To illustrate further the difference in simulated amines concentrations for SDR and FR cases and the effect of wind directions, we present in Fig. 10 the horizontal distributions of simulated C2-amines concentrations in domain 3 (horizontal resolution 9 km×9 km) at 18:00 on 26 August and at 2:00 on 29 July 2015. As shown in Fig. 5d, the observed C2-amines concentration at the NUIST site at 18:00 on 26 August is ~19 pptv, and the corresponding



simulated value is slightly lower based on SDR (13.4 pptv) while it is significantly lower (by a factor of ~8) based on FR (2.5 pptv). Similar difference can also be seen for the Fudan site at 2:00 on 29 July 2015 (Fig. 6d). It can be seen from Fig. 10b and d that the significantly lower predicted concentrations of C2-amines are not limited to the NUIST and Fudan sites, but for

the whole region. It is noteworthy that C2-amines concentrations downwind of heavy industrial zones (north-east of the NUIST site) (Fig. 10a) is reproduced well, indicating the contribution of industrial sources to C2-amines concentrations observed at the NUIST site. For the Fudan site, residential contribution from the highly populated urban center is essential to maintain the relatively higher C2-amines concentrations.

As we show in this section, the results based on SDR are overall in much better agreement with measurements than those based on FR assumed in previous studies. Nevertheless, there still exist large differences between SDR simulations and observations (Figs. 5-9). The differences can be caused by many factors including, but not limited to, uncertainties in emission inventories (both ammonia and the derived amines to ammonia ratios), meteorology,

oxidation and aerosol uptake of amines, and measurements. Further research is needed to reduce these uncertainties.

### 3.3 Spatial distribution of methylamines over Yangtze River Delta

Figure 11 presents simulated mean (25 July – 25 August) surface layer horizontal distributions of mean C1-, C2-, and C3-amines for the SDR2015 case in the Yangtze River

Delta region (left panels) and the Shanghai area (right panels). It can be clearly seen that high methylamines concentrations are typically confined to source regions, with very low concentrations over oceans. Figs. 11a, c, and e show that averaged C1-, C2-, and C3-amines concentrations in the surface layer in Yangtze River Delta region (based on domain 9 km×9 km resolution results) are, respectively, in the range of 2-20 pptv, 5-50 pptv, and 0.5-4 pptv,

with spatial pattern similar to that of emissions (Figs. 2-4f). C2-amines concentrations in urban areas are higher than those in agricultural areas, while C1-amine and C3-amines concentrations show high values in agricultural areas such as in Zhejiang province, except for areas of high



urbanization. Further measurements in these regimes of high concentrations are needed to constrain the model simulations. Considering the complex underlying surface in urban Shanghai, we apply 4-domain-nested simulations to further study the Shanghai urban area. As shown in section 3.2, simulations with higher spatial resolution are in better agreement with

measurements. Figures 11b, d, and f, which were based on domain 4 simulations (3 km×3 km resolution), show that the Shanghai urban area are hot-spots for C1-, C2-, and C3-amines, with concentrations in Shanghai dowantown up to ~15 pptv, 50 pptv, and 4 pptv, respectively. It can be seen clearly that Fudan site is on the edge of the central area such that methylamines concentrations are affected easily downwind of the city center, especially for C2-amines.

Vertically, the concentrations of C1-, C2, and C3-amines decrease quickly with altitude (Fig. 12), dropping by a factor of ~10 from the surface to ~900 hPa. The horizontal and vertical distributions of methylamines for the SDR2012 case are similar to that for the SDR2015 case and are not shown. The fact that the high concentrations of methylamines are confined to source regions and the boundary layer are as a result of their short life time, again highlighting

the necessity to better quantify the emissions of amines from different sources and to model with high spatial resolutions to study their spatial distributions and potential impacts.

## 4. Summary and discussion

A few pptv of gaseous amines have been observed to be able to significantly enhance new particle formation in the atmosphere (Almeida et al., 2013; Chen et al., 2014; Jen et al., 2016,

Lehtipalo et al., 2016). Recent field measurements (Zheng et al., 2015; Yao et al., 2016) indicate that gaseous amines in the Yangtze River Delta region, China can reach a few tens pptv with large temporal variations. To understand the processes controlling the concentrations of amines and their spatio-temporal distribution in the atmosphere, we improve a previous method in estimating amines emissions by distinguishing amines emissions from five different

source types and simulating amines concentrations over the Yangtze River Delta with a regional model (WRF-Chem).

The present study calculates methylamines emissions from five source types, including chemical industry, other industry, agriculture, residential, and transportation. The temporal and



spatial variations of methylamines emissions are assumed to follow that of ammonia for different sources. The amines-to-ammonia mass emission ratios, derived from previous measurements reported in Zheng et al. (2015), are 0.026, 0.0015, 0.0011, 0.0011, and 0.0011 for C1-amine, 0.007, 0.0018, 0.0015, 0.01, and 0.0009 for C2-amines, and 0.0004, 0.0005,

0.00043, 0.0006, and 0.0004 for C3-amines for chemical-industrial, other industrial, agricultural, residential, and transportational sources, respectively. Ammonia, C1-, C2-, and C3-amines emission flux in Yangtze River Delta are 919.61 Gg N yr$^{-1}$, 551.88, 849.11, and 117.78 Mg N yr$^{-1}$, respectively. The contributions of chemical-industrial, other industrial, agricultural, residential, and transportational sources to domain-average methylamines (C1-

amine+ C2-amines+ C3-amines) are 0.73%, 0.81%, 66.04%, 30.81%, and 1.61%, respectively, which shows that agricultural and residential source types dominate methylamines emissions over the Yangtze River Delta.

Three tracers representing C1-, C2-, and C3-amines have been added into WRF-Chem and simulations with multiple nested domains have been carried out. The simulated concentrations

of C1-, C2-, and C3-amines, based on fixed ratios (FR) of amines to ammonia assumed in previous studies and source dependent ratios (SDR) derived in the present study, have been compared with field measurements at a suburban site in Nanjing, China and at an urban site in Shanghai, China. We show that SDR substantially improves the ability of the model in capturing the observed concentrations of methylamines. C1-, C2-, and C3-amines

concentrations are in the range of 2-20 pptv, 5-50 pptv, and 0.5-4 pptv in the surface layer in the Yangtze River Delta region. Vertically, the concentrations of C1-, C2-, and C3-amines decrease by a factor of ~10 from the surface to ~900 hPa. High concentrations of methylamines are generally confined to source regions and the boundary layer as a result of their short life time. For the urban Fudan site, simulated concentrations downwind of areas of high residential

activities are closer to site measurements than for other wind directions, suggesting that residential sources are important in an urban area and that the present estimation of residential emissions may be reasonable.

It should be pointed out that the uncertainties in emissions (of both ammonia and amines to ammonia ratios), meteorology, aerosol uptake, and chemical processes can all impact the





simulated values of amines in this study. To advance the accuracy of amines emissions, more field observations as well as more accurate source apportionment of amines are needed. This study focuses on the summer season due to limited measurements, but the model approach developed here can be applied to study the seasonal characteristics of methylamines and

5 subsequently the impact of amines on new particle formation and growth in the future.

Acknowledgements: This work was financially supported by the National Key Research and Development Program of China (Grant No. 2016YFA060130X), the National Natural Science Foundation of China (21677083, 91644213), and the National Science Foundation of US

(1550816). LW thanks the Royal Society Newton Advanced Fellowship (NA140106). JZ thanks the funding from NSFC 41275142 and 91644213.

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

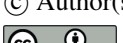


Table 1. The ratios of C1-, C2-, and C3-amines concentrations of that of ammonia from individual plumes with different origins, derived from measurements taken at a suburban site of Nanjing as reported in Zheng et al. (2015).

| Plume # | Time | [C1-amine]/ [NH$_3$] | [C2-amine]/ [NH$_3$] | [C3-amine]/ [NH$_3$] | Source type identified |
|---|---|---|---|---|---|
| 1 | ~22:00 8/26 | 0.0010 | 0.0018 | 0.0002 | |
| 2 | ~20:00 8/29 | 0.0009 | 0.0012 | 0.0004 | other industry |
| 3 | ~10:00 8/30 | 0.0009 | 0.0024 | 0.0008 | except for |
| 4 | ~17:00 8/30 | 0.0013 | 0.0018 | 0.0006 | chemistry |
| 5 | ~14:00 8/31 | 0.0032 | 0.0018 | 0.0005 | |
| 6 | ~9:00 8/28 | 0.0010 | 0.0015 | 0.0003 | agriculture |
| 7 | ~14:00 8/28 | 0.0012 | 0.0014 | 0.0006 | |
| 8 | ~8:00 8/29 | 0.0011 | 0.0009 | 0.0004 | transportation |
| 9 | ~21:00 8/27 | 0.0011 | 0.0100 | 0.0006 | residential |

Table 2. Simulation cases in the study.

| Case | Methods of calculating Methylamines emissions | Simulation Periods |
|---|---|---|
| FR2012 | Fixed Ratios (Schade and Crutzen, 1995) | Aug 26-Aug 31, 2012 |
| SDR2012 | Source-dependent Ratios (This study) | |
| FR2015 | Fixed Ratios (Schade and Crutzen, 1995) | Jul 25-Aug 25, 2015 |
| SDR2015 | Source-dependent Ratios (This study) | |



Table 3. Emission rates of ammonia, C1, C2, C3-amines from different sources based on SDR for domain 3.

|  | Ammonia | C1-amine | C2-amines | C3-amines |
|---|---|---|---|---|
| agariculture | 785.20 | 460.73 | 444.94 | 97.28 |
| residential | 103.09 | 62.19 | 389.47 | 16.34 |
| transportation | 23.19 | 13.48 | 7.88 | 3.01 |
| other industry | 7.47 | 6.15 | 5.08 | 1.08 |
| chemical industry | 0.65 | 9.32 | 1.73 | 0.08 |
| Total | 919.61 | 551.88 | 849.11 | 117.78 |

Notes: the unit of ammonia: Gg (N) yr[-1], the unit of C1-, C2-, andC3-amines: Mg (N) yr[-1]



Table 4. Statistical performance methylamines simulation at NUIST site (FR2012, SDR2012) in domain 3 and Fudan site (FR2015, SDR2015) in both domain 3 and domain 4 (values given in parentheses).

| Case | Variable | No.samples | Obs.ave | Sim.ave (Domain4) | NMB (Domain4) |
|---|---|---|---|---|---|
| NUIST FR2012 | C1-amine | 61 | 4.35 | 8.97 | 106.72 |
| | C2-amines | 61 | 7.08 | 1.99 | -71.50 |
| | C3-amines | 61 | 1.91 | 8.64 | 359.02 |
| NUIST SDR2012 | C1-amine | 61 | 4.35 | 6.39 | 45.60 |
| | C2-amines | 61 | 7.08 | 10.56 | 49.12 |
| | C3-amines | 61 | 1.91 | 1.12 | -41.26 |
| Fudan FR2015 | C1-amine | 719 | 15.71 | 6.79 (9.26) | -56.75 (-41.03) |
| | C2-amines | 719 | 40.20 | 1.56 (2.15) | -96.13 (-94.67) |
| | C3-amines | 719 | 1.13 | 6.71 (9.24) | 494.28 (718.61) |
| Fudan SDR2015 | C1-amine | 719 | 15.71 | 4.97 (6.61) | -68.37 (-57.95) |
| | C2-amines | 719 | 40.20 | 16.33 (25.15) | -59.37 (-37.43) |
| | C3-amines | 719 | 1.13 | 1.01 (1.37) | -10.84 (21.34) |

Notes: the unit of Obs.ave and Sim.ave: pptv, the unit of NMB: %





Table 5. Variations in normalized mean bias (NMBs) of methylamines simulations when amines emission rates are halved or doubled, at NUIST site (SDR2012, 0.5 SDR2012, 2 SDR2012) in domain 3 and Fudan site (SDR2015, 0.5 SDR2015, 2 SDR2015) in domain 4.

| Sensitivity Case | C1-amine | C2-amines | C3-amines |
|---|---|---|---|
| SDR2012 | 45.60 | 49.12 | -41.26 |
| 0.5 SDR2012 | -27.96 | -25.73 | -74.31 |
| 2 SDR2012 | 193.13 | 199.01 | 88.97 |
| SDR2015 | -51.23 | -9.73 | 69.62 |
| 0.5 SDR2015 | -75.99 | -55.10 | -28.51 |
| 2 SDR2015 | -2.00 | 80.34 | 513.09 |





**Figure**

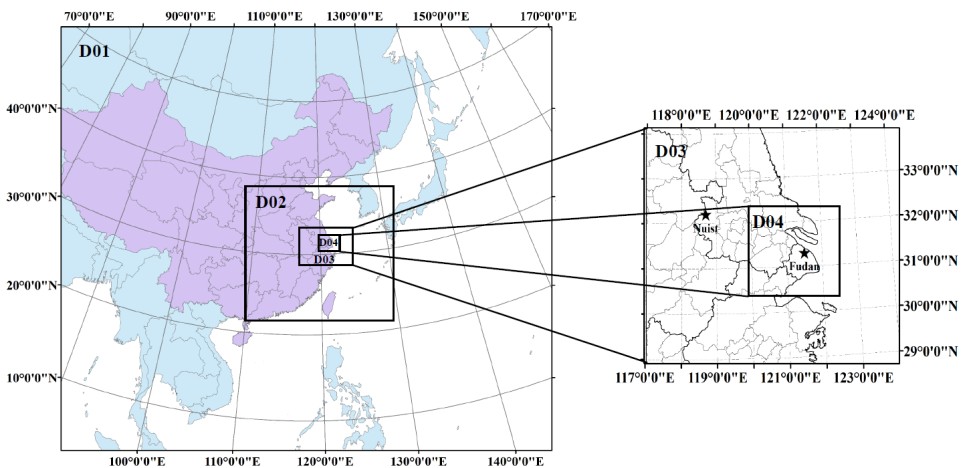

Figure 1. Four nested domains in the present study. Domain 1 covers East-Asia and part of south-east Asia. Nested domain 2, 3, and 4 cover a large part of East- China, the Yangtze River
5    Delta (including Nanjing, Shanghai), and Shanghai, respectively.





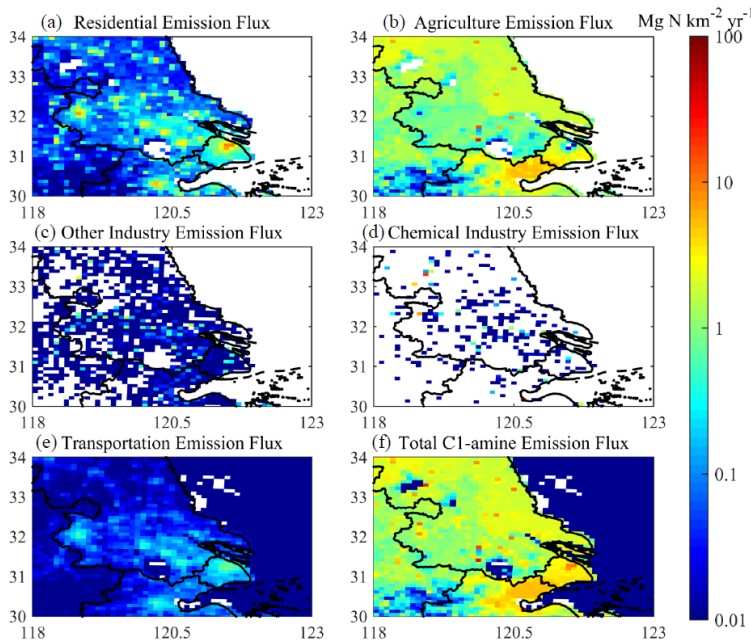

Figure 2. The horizontal emission flux distributions for C1-amines: (a) residential; (b) agriculture; (c) other industry; (d) chemical industry; (e) transportation; (f) total.





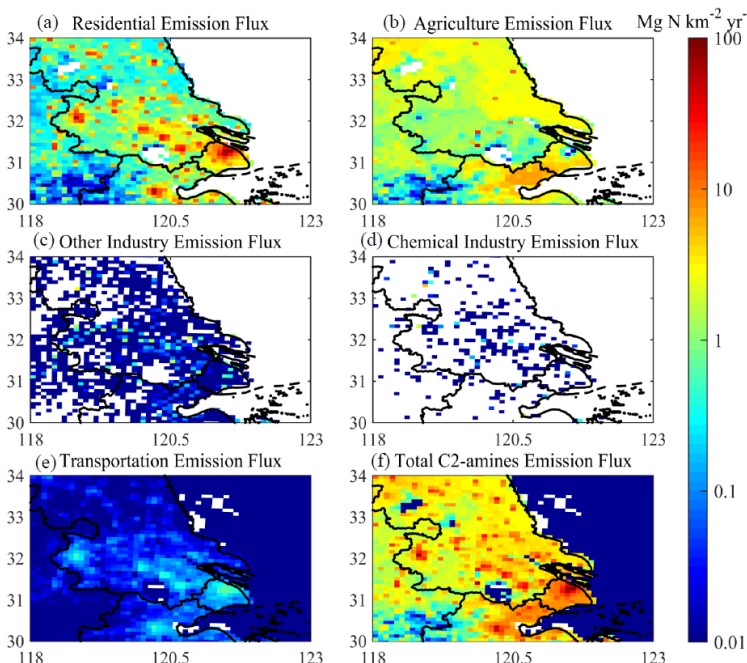

Figure 3. Same as Fig. 2 but for C2-amines.





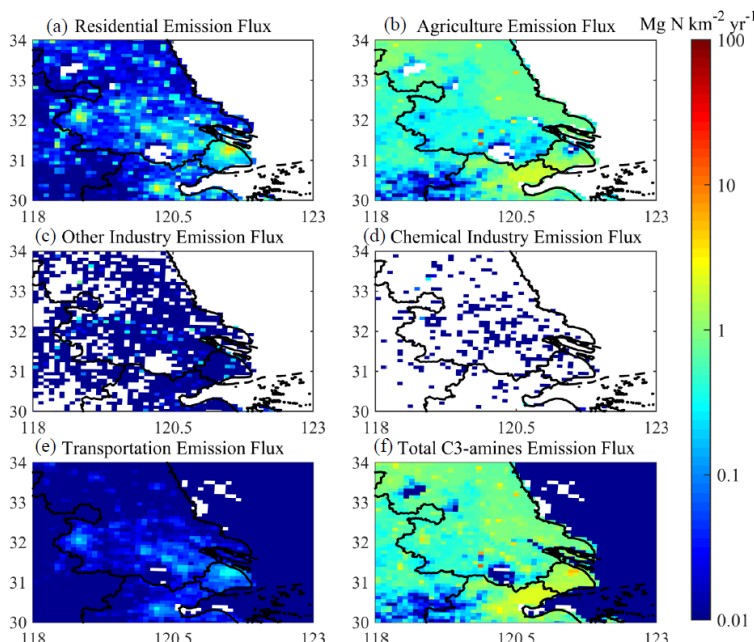

Figure 4. Same as Fig. 2 but for C3-amines.





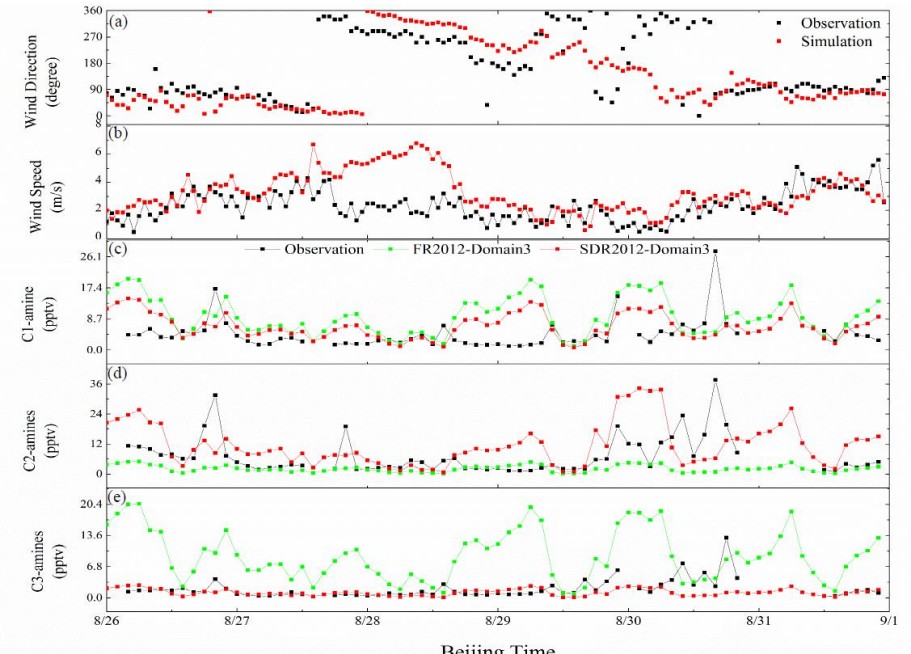

Figure 5. Comparisons of simulated and observed wind direction at 10 m (a), wind speed at 10 m (b), C1-amine (c), C2-amines (d), C3-amines (e) concentrations at the NUIST site in Nanjing, China from 26 August to 31 August, 2012. In Figs. 5c-e, black, red, and green lines represent observations, simulated values in domain3 based on SDR, and simulated values in domain3 based on FR, respectively.

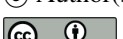



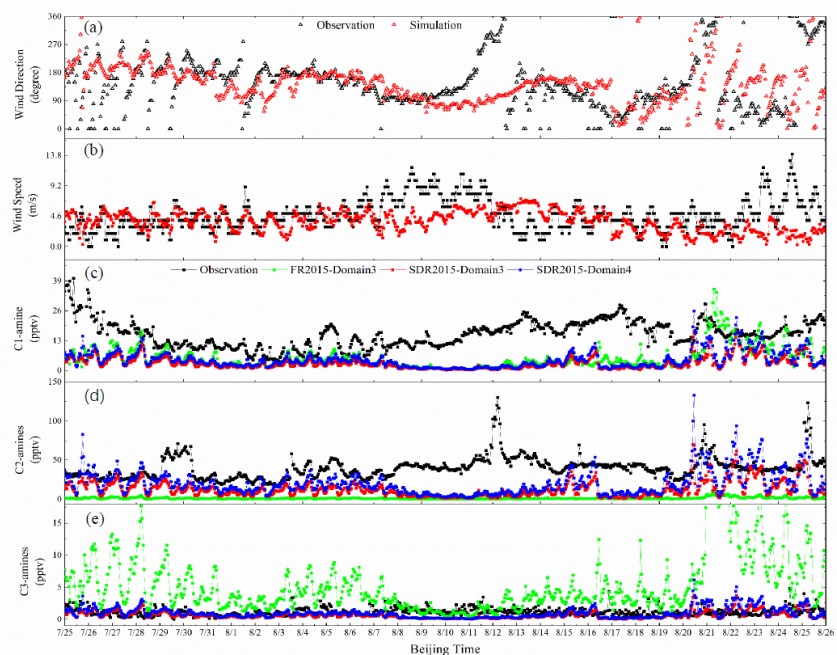

Figure 6. Comparisons of simulated and observed wind direction (a), wind speed (b), C1-amine (c), C2-amines (d), C3-amines (e) concentrations at the Fudan site in Shanghai, China from 25 July to 25 August 2015. In Figs.6c-e, black, blue, red, and green lines present observations, simulated values in domain 4 based SDR, domain 3 based on SDR, and domain 3 based on FR, respectively.





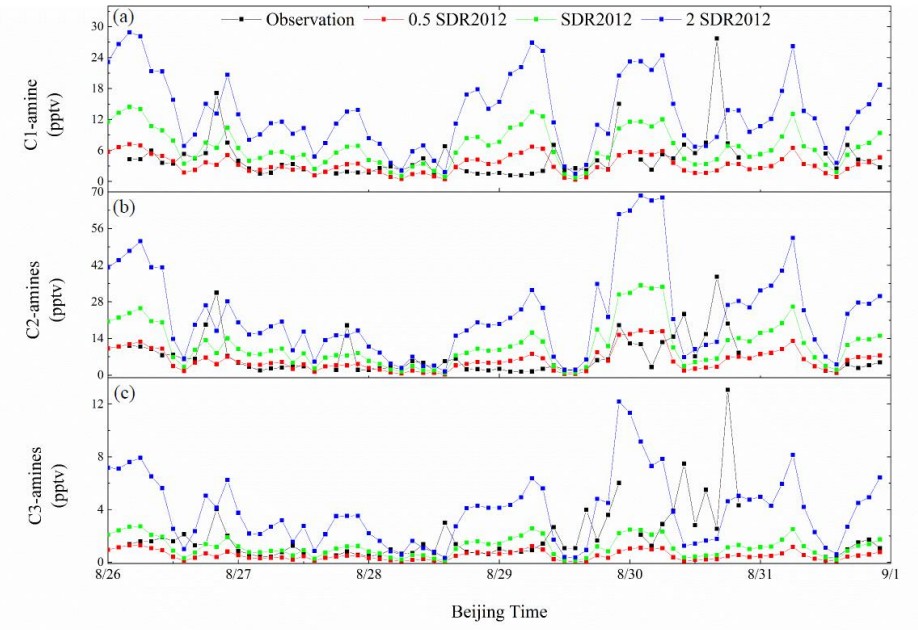

Figure 7. Comparisons of simulated C1-, C2-, C3-amines concentrations in SDR2012, 0.5
SDR2012, and 2 SDR2012 (domain 3) at the NUIST site with measurements from 26 August
to 31 August, 2012 . Black, red, green, and blue lines present observations, simulated values
in 0.5 SDR2012, SDR2012, and 2 SDR2012, respectively.





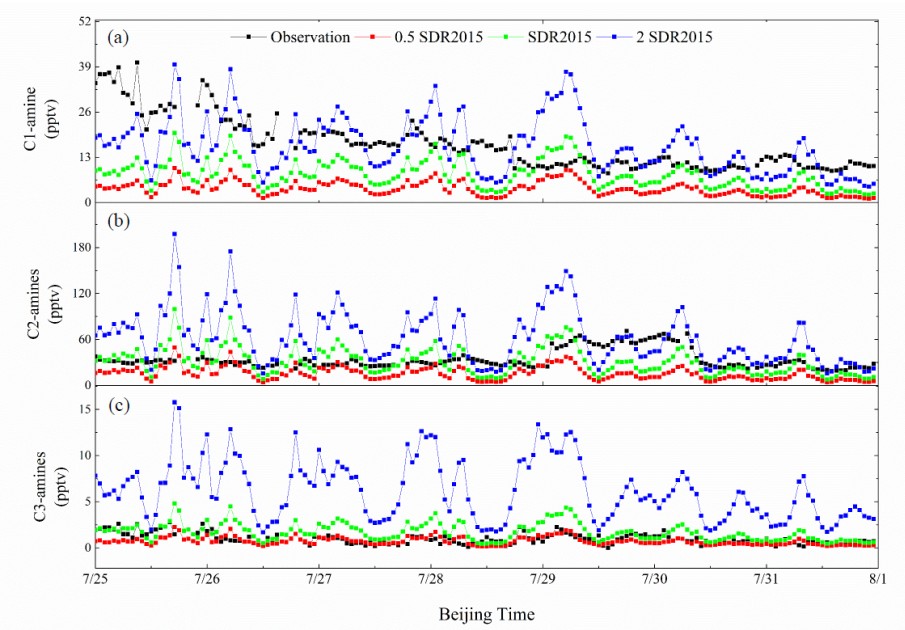

Figure 8. Same as Fig. 7 but for 0.5 SDR2015, SDR2015, and 2 SDR2015 (domain 4) at the
Fudan site from 25 July to 31 July, 2015.



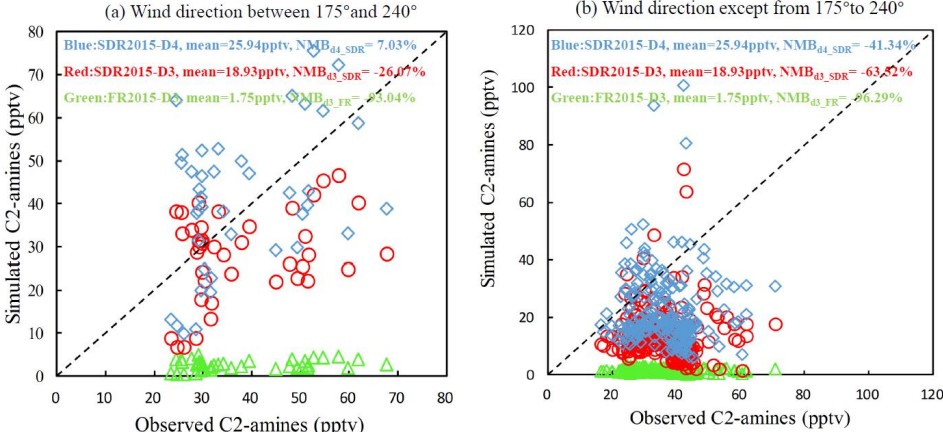

Figure 9. Comparisons of simulated and observed C2-amines at the Fudan site for different wind direction zones: (a) Wind directions between 175° and 240° when the Fudan site is downwind of high residential emissions; (b) Other wind directions. $NMB_{d3\_SDR}$, $NMB_{d4\_SDR}$, and $NMB_{d3\_FR}$ represent normalized mean bias in domain 3 and domain 4 based on SDR and domain 3 based on FR, respectively.





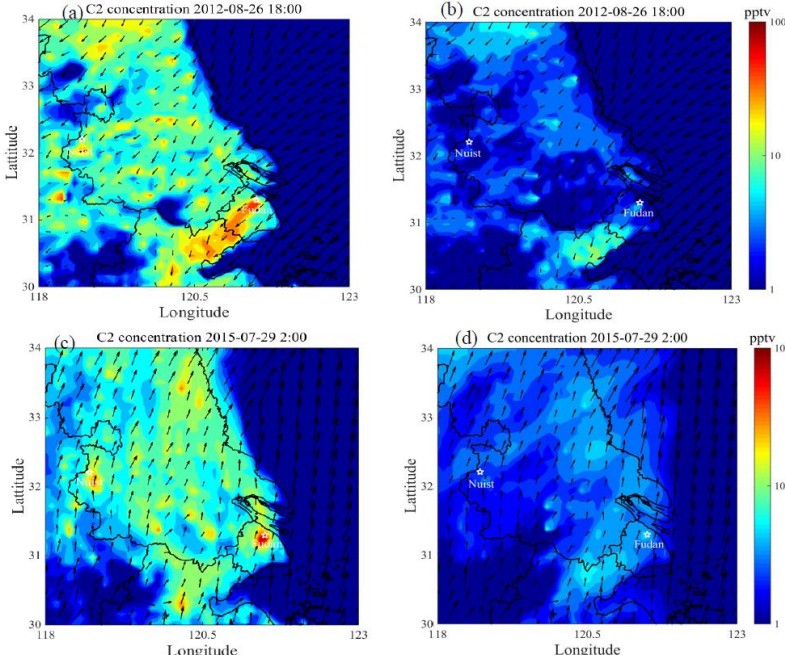

Figure 10. Simulated C2-amines concentrations at 18:00 on 26 August 2012 and 2:00 on 29 July 2015 using SDR (a, c) and FR (b, d) in domain 3 (horizontal resolution at 9 km×9 km).





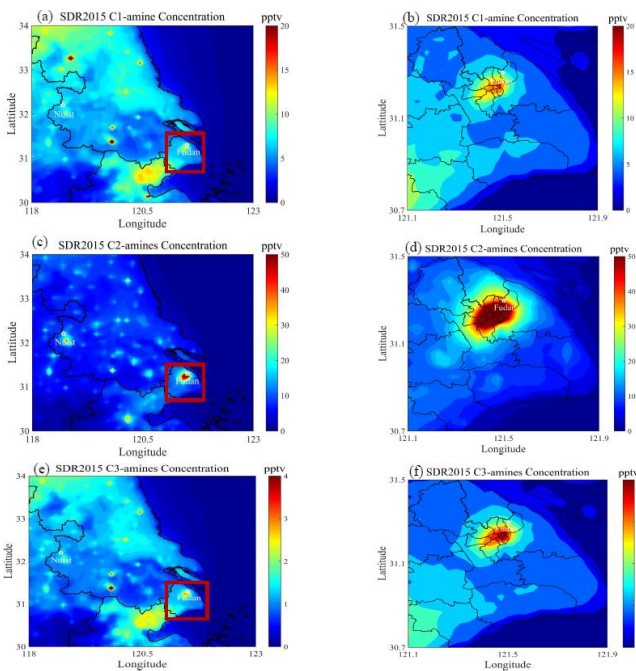

Figure 11. Simulated horizontal distributions of mean concentrations of C1-, C2-, and C3-amines in domain 3 and domain 4 during the period of 25 July to 25 August 2015 (SDR2015 case).





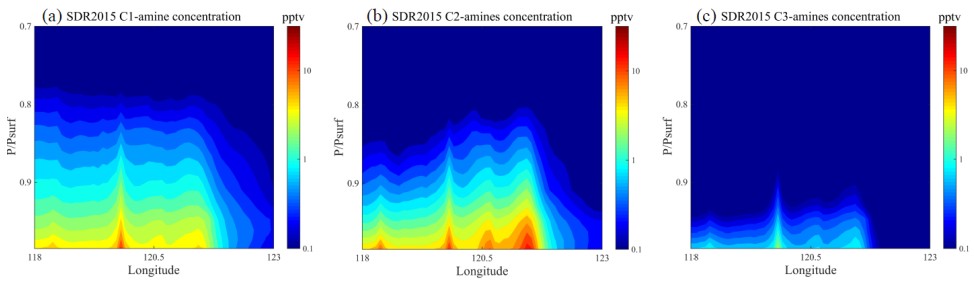

Figure 12. Simulated vertical distributions of mean concentrations of C1-, C2-, and C3-amines in domain 3 during the period of 25 July to 25 August 2015 (SDR2015 case).