# Peer review of "High resolution modeling of gaseous methylamines over a polluted region in China: Source-dependent emissions and implications to spatial variations"

_Atmospheric Chemistry and Physics, 2017_

## Referee Comment (RC1) · Anonymous Referee #1 · 27 Jan 2018

Gaseous amines represent a category of base compounds which plays import roles in many aspects of atmospheric chemistry including nucleation and growth of newly-formed particles. Compared to ammonia, concentrations of individual amines are several orders of magnitude lower, far below ppb levels. In addition, there are a variety of sources of amines in the atmosphere. Furthermore, most amines are rather reactive, bearing shorter lifetimes than ammonia. Hence the temporal and spatial distributions of amines can vary significantly. This paper presents a high resolution modeling study of methylamines (C1-C3) in Yangtze River Delta Region (YRD) by considering

source dependent amine-to-ammonia ratios (SDR) whose results demonstrate much better agreement with observations than those assuming fixed ratios (FR) in the model simulations. Here four domains are considered and the simulated results from the smallest two domains showed that models with higher spatial resolution yield better agreement with observations, demonstrating the need for employing high resolutions when modeling spatial distributions of amines in order to better understand their roles in atmospheric chemistry. The paper can be publishable after the following issues are resolved: 1. The paper models the amine concentrations and their spatial distributions from five different source types (chemical industry, other industry, agriculture, residential, and transportation). What is the rationale behind this classification? Are there any previous studies that employed a similar classification? 2. The study used measured data from two sites (NUIST and Fudan sites). Since the measured amine concentrations might be strongly affected by the close-to-site sources, the authors should provide some evidences that those sites are not significantly affected by local sources which may lead to systematic biases for the data. According to Table 4, the Fudan site may be affected significantly by local sources. 3. Table 3 lists emission rates of C1-C3 amines from different sources based on the SDR ratios from this study. However, it is not very clear how those values are obtained. In section 2.2, the authors only used SDR from the data measured in 2012 (NUIST site) and did not even mentioned those measured in 2015 (Fudan site). The authors should provide the reasons for only considering one data set rather than both data sets. In addition, the paper mentioned very briefly the uncertainties associated with the measured data. Can those uncertainties be quantified? How a single (or even two) measured site can be representative of the domains of interest (i.e., D3 and D4)? How those five different sources of amines are determined, for example, based on what criteria, the emission rates of the five sources are distributed? 4. Some rather minor points: 1) L7 on p2, change "model's" to "of the model"; similarly for "model's skill" on p7 (L27); 2) L27 on p4, change "amines concentrations" to "concentrations of amines"; there are lots of those usages throughout the paper. Please correct them; 3) L1-2 on p5, year 2014 is not up-to-date; 4) L23 on p5,

change "The point sources data" to "the data of the point sources"; 5) L15 on p6, "at an urban site" not "in an urban site"; 6) L21 on p6, delete "seek to"; 7) L9 on p7, "in details" not "in detail"; 8) L15-20 on p7, this ratio of 0.026 might be problematic if the measured site is so close to the source and affected strongly by the emissions from the source; 9) L23 on p7, delete "would like to"; 10) L4 on p8, "prior to this study" might be better replaced by "in previous studies"; 11) L9 on p10, change "that" to "those" since it refers to as "distributions"; 12) L28 on p10, "general underprediction of the model", do you mean that it is compared to measurements? 13) L10-11 on p11, where those values are from? 14) L18-20 on p11, I don't think wind direction and speed are the reasons.
* * *

---

## Referee Comment (RC2) · Anonymous Referee #2 · 5 Feb 2018

Gas-phase amines can influence the new particle formation and growth in the atmosphere. Although their concentrations in the ambient air are clearly lower than ammonia they play an important role in the particle formation and growth due to higher reactivity compared ammonia. Largely due to the lower concentrations and higher reactivity they will only affect the processes near the source regions.

Due to lack measurements of amines previously the emissions of amines have been modelled using fixed ratios (FR) between ammonia and amines. This paper presents a simulation study over the Yangtze River Delta Region to produce and test source

dependent amine-to-ammonia ratios (SDR) in order to improve future model simulations of amines in the atmosphere. The idea is worthy and can produce a significant contribution to the field. However, there several things that need to be improved before publication. In the following I detail the changes by sections that are needed before publication:

**Methods**

- please state the emission frequency (daily, hourly, more frequent?), what is available in the dataset and what is used in this study.
- Is the emission data available online, and/or how to get it?

- What is the reasoning behind the emission sectors?
- In Zheng et al. (2015) and current study, the times for observations are different, why? I don't see any other than "other industry" sector in Zheng et al. (2015), where are the other emission factors coming? The numbers do not match with Zheng et al. (e.g. 31.8. [C1/NOx]/[NH3/NOx]:0.000076/0.037=0.0021 and current works states 0.0032) or am I missunderstanding something? And please describe the calculation in the text.
- SDR is based on NUIST, but main study on Fudan, why not do two simulations with the finest resolution for both stations?

- Model description must be improved, now the authors only say they follow Yu & Luo (2014), but this is the first time of implementing amine compounds in WRF-Chem, it needs to be explained in detail

Absolutely necessary information:
- What is the particle uptake mechanism for amines?
- What are the oxidation coefficients? and which oxidants?

- other removal mechanisms? wet depostion for example?

**Results**

- NMBs in Table 4 are not correct, it looks like that they are only bias of the total mean ($\frac{(\overline{C_m} - \overline{C_o})}{\overline{C_o}}$). Correct way to calculate NMB is $\frac{\sum_{i=1}^{N}(C_m - C_o)}{\sum_{i=1}^{N} C_o}$, check Boylan & Russell (2006) for more information. As it is now, it can give a wrong impression of model ability to reproduce observations.

- It would be reasonable to focus on NUIST since the emission (SDR) factors are based on this station, so could you make run with the domain4 also for NUIST
- Please add domain 3 for Fudan in Table 5 also, to facilitate comparing to NUIST site
- Please analyse the discrepancy between model and observations more carefully, now the reasons for discrepancies are vague
- in addition to separating Fudan by agricultural/residential sector, add separation by land/sea also. This would allow evaluating non-pollution sector concentrations.
- The sensitivity test is doubling/halving SDRs only. Can you use the uncertainty from observations to create uncertainty range in SDR and do sensitivity test with max/min range for that, could you do different particle uptake coefficients, this would be interesting. This way we could have an idea to which of the uncertainties are in most urgent need of new research

- Authors refer to short lifetime for amine many times without a reference or calculation of lifetime of amines, please add reference and/or calculation from your model
- Can you compare the particle size distributions with observations to evaluate the particle sink for amine?

**Boylan, J.W., Russell, A.G., 2006.** PM and light extinction model performance metrics, goals, and criteria for three- dimensional air quality models. Atmospheric Environment 40, 4946-4959.
* * *

---

## Author Response (AR1)

We would like to thank both reviewers for their constructive comments which help to improve our manuscript. Our point-to-point replies (in blue) to the comments are given below (the original comments are copied here in black). The manuscript has been revised accordingly. All the changes to the manuscript have been highlighted using the Microsoft word "track-changes" tool in one version of the submitted revised manuscript.

**Anonymous Referee #1**

Gaseous amines represent a category of base compounds which plays import roles in many aspects of atmospheric chemistry including nucleation and growth of newly formed particles. Compared to ammonia, concentrations of individual amines are several orders of magnitude lower, far below ppb levels. In addition, there are a variety of sources of amines in the atmosphere. Furthermore, most amines are rather reactive, bearing shorter lifetimes than ammonia. Hence the temporal and spatial distributions of amines can vary significantly. This paper presents a high resolution modeling study of methylamines (C1-C3) in Yangtze River Delta Region (YRD) by considering source dependent amine-to-ammonia ratios (SDR) whose results demonstrate much better agreement with observations than those assuming fixed ratios (FR) in the model simulations. Here four domains are considered and the simulated results from the smallest two domains showed that models with higher spatial resolution yield better agreement with observations, demonstrating the need for employing high resolutions when modeling spatial distributions of amines in order to better understand their roles in atmospheric chemistry.

Thanks for the nice summary and the positive comments.

The paper can be publishable after the following issues are resolved:

1. The paper models the amine concentrations and their spatial distributions from five different source types (chemical industry, other industry, agriculture, residential, and transportation). What is the rationale behind this classification? Are there any previous studies that employed a similar classification?

In various emission inventories such as MEIC and INTEX-B, emission sources are generally

separated into different types like residential, agriculture, transportation, industry, and power plant. Considering that emission rates of amines from organic synthesis may differ significantly with those from power generation and heavy industries using selective catalytic reduction (Zheng et al., 2015), we divided industrial sources into chemical industry and other industry in the present study. As emphasized in the Introduction, no previous modeling studies (to our knowledge) have distinguished different amines emission from various source types.

2. The study used measured data from two sites (NUIST and Fudan sites). Since the measured amine concentrations might be strongly affected by the close-to-site sources, the authors should provide some evidences that those sites are not significantly affected by local sources which may lead to systematic biases for the data. According to Table 4, the Fudan site may be affected significantly by local sources.

In this study we used amines to ammonia ratios in various plumes observed at the NUIST site to derive source-dependent amines emissions. Ammonia was not measured at the Fudan site during the period when amines were measured. We agree that concentrations of amines can be strongly affected by the close-to-site sources. Nevertheless, we do not use absolute concentration of amines, but the ratios of amines to ammonia to derive amine emissions. Therefore, the effect of local sources does not impact the conclusions of this paper.

3. Table 3 lists emission rates of C1-C3 amines from different sources based on the SDR ratios from this study. However, it is not very clear how those values are obtained. In section 2.2, the authors only used SDR from the data measured in 2012 (NUIST site) and did not even mentioned those measured in 2015 (Fudan site). The authors should provide the reasons for only considering one data set rather than both data sets. In addition, the paper mentioned very briefly the uncertainties associated with the measured data. Can those uncertainties be quantified? How a single (or even two) measured site can be representative of the domains of interest (i.e., D3 and D4)? How those five different sources of amines are determined, for example, based on what criteria, the emission rates of the five sources are distributed?

We derived emission rates of C1-C3 amines listed in Table 3 based on SDR ratios and ammonia

emissions from different sources. We have refined relevant description in the manuscript to make this clearer.

We only used SDR from the data measured in 2012 (NUIST site) because only the NUIST data has simultaneous measurements of NH3, NOx, and SO2 along with amines, enabling us to identify the plume source types (as detailed in Zheng et al., 2015). These species were not measured at the Fudan site during the period when amines were measured.

As emphasized in the manuscript, the estimation of amines emissions from different sources is subject to a large uncertainty, mainly due to very limited measurements available to constrain the estimation. We agree with the reviewer's concern about the representativeness of limited measurements and this can only be resolved by more similar measurements. The present study is the first attempt (to our knowledge) to use direct measurements to constrain amine emissions from different sources. The SDR approach, as we show here, improves the skill of the model in simulating concentrations of amines in polluted regions. We hope more field observations as well as more accurate source apportionment of amines will be carried out in the future to constrain the amines emissions and model study.

We derived regional methylamines emissions based on amines to ammonia ratios and ammonia emissions from different sources. The temporal and spatial distributions of C1-, C2-, and C3-amines follow those of ammonia for different sources.

**4. Some rather minor points:**

 L7 on p2, change "model's" to "of the model"; similarly for "model's skill" on p7 (L27) Modified.

 L27 on p4, change "amines concentrations" to "concentrations of amines"; there are lots of those usages throughout the paper. Please correct them;
 Modified.

3) L1-2 on p5, year 2014 is not up-to-date;

We mean the emission inventory for the year 2014 is up-to-date. To avoid confusion, we have

deleted "up-to-date" from the sentence.

4) L23 on p5, C2 change "The point sources data" to "the data of the point sources";

Modified.

5) L15 on p6, "at an urban site" not "in an urban site";

Modified.

6) L21 on p6, delete "seek to";

Modified.

7) L9 on p7, "in details" not "in detail";

Modified.

8) L15-20 on p7, this ratio of 0.026 might be problematic if the measured site is so close to the

source and affected strongly by the emissions from the source;

The ratio of 0.026 is detected in the ammonia water from a local chemical supplier.

9) L23 on p7, delete "would like to";

Modified.

10) L4 on p8, "prior to this study" might be better replaced by "in previous studies";

Modified.

11) L9 on p10, change "that" to "those" since it refers to as "distributions";

**Modified.**

12) L28 on p10, "general underprediction of the model", do you mean that it is compared to measurements?

Yes.

13) L10-11 on p11, where those values are from?

We derived the values based on Fig.5-6 and Table 4.

14) L18-20 on p11, I don't think wind direction and speed are the reasons.

It is noted that many reasons may cause the underestimation, but at least partially due to the large deviation of the simulated wind directions and speeds during the period.

**Anonymous Referee #2**

Gas-phase amines can influence the new particle formation and growth in the atmosphere. Although their concentrations in the ambient air are clearly lower than ammonia they play an important role in the particle formation and growth due to higher reactivity compared ammonia. Largely due to the lower concentrations and higher reactivity they will only affect the processes near the source regions. Due to lack measurements of amines previously the emissions of amines have been modelled using fixed ratios (FR) between ammonia and amines. This paper presents a simulation study over the Yangtze River Delta Region to produce and test source dependent amineto-ammonia ratios (SDR) in order to improve future model simulations of amines in the atmosphere. The idea is worthy and can produce a significant contribution to the field.

We appreciate the positive comments confirming the importance of this study.

However, there several things that need to be improved before publication. In the following I detail the changes by sections that are needed before publication.

Methods

- please state the emission frequency (daily, hourly, more frequent?), what is available in the dataset and what is used in this study.

The emission frequency in present study is hourly with a daily cycle. The dataset contains emissions of five source types: residential, agriculture, transportation, chemical industry, and other industry. This has been clarified in the revised manuscript.

**- Is the emission data available online, and/or how to get it?**

We derived methylamines emissions based on ammonia emissions for different sources, causing the emission frequency is the same as ammonia. In our study, for ammonia emissions, the dataset for MEIC is available online, and we download it from http://www.meicmodel.org, while we get the refined emission data for ammonia in YRD from the Shanghai Academy of Environmental Sciences (SAES). Interested reader can use the amines to ammonia ratios presented in this manuscript to calculate amines emissions.

- What is the reasoning behind the emission sectors?

In various emission inventories such as MEIC and INTEX-B, emission sources are generally separated into different types like residential, agriculture, transportation, industry, and power. Considering that emission rates of amines from organic synthesis may differ significantly with those from power generation and heavy industries using selective catalytic reduction (Zheng et al., 2015), we divided industrial sources into chemical industry and other industry in the present study.

- In Zheng et al. (2015) and current study, the times for observations are different, why? I don't see any other than "other industry" sector in Zheng et al. (2015), where are the other emission factors The coming? numbers do not match with Zheng et al. (e.g. 31.8. [C1/NOx]/[NH3/NOx]:0.000076/0.037=0.0021 and current works states 0.0032) or am I missunderstanding something? And please describe the calculation in the text.

We chose a one-week period (26 August to 31 August 2012) instead of all the observation period because plumes with high concentrations of amines and ammonia were measured only during this period.

For the source types: residential, agriculture, transportation, and other industry, we derived the ratios according to the peak values of amines and ammonia in the plumes identified by and as shown in Fig. 6 of Zheng et al. (2015). As described in the manuscript, the source of ratio for the chemical industry is based on the direct measurement of amines in the ammonia water solution used as absorbent during flue gas treatment.

Numbers in our manuscript are peak values of plumes shown in Fig.6 of Zheng et al. (2015) while the ratios of NH3 and amines to NOx given in Table 2 of Zheng et al. (2015) (for five industrial plumes only) were acquired by using orthogonal distance regression analyses. The table below shows the ratios in the five plumes in Zheng et al. and present study, respectively. We used the averaged peak values of five plumes in present study. The slight difference in the ratios does not affect the main conclusions of this study.

| Reference     | Plume # | C1/NH3  | C2/NH3 | C3/NH3 |
|---------------|---------|---------|--------|--------|
|               | 1       | 0.0011  | 0.0015 | 0.0002 |
|               | 2       | 0.0008  | 0.0008 | 0.0002 |
|               | 2       | 0.0008  | 0.0023 | 0.0008 |
| Zheng et al.  | 4       | 0.0015  | 0.0018 | 0.0005 |
|               | 5       | 0.0021  | 0.0011 | 0.0017 |
|               | average | 0.0013  | 0.0015 | 0.0007 |
|               | 1       | 0.001   | 0.0018 | 0.0002 |
|               | 2       | 0.0009  | 0.0012 | 0.0004 |
|               | 3       | 0.0009  | 0.0024 | 0.0008 |
| Present study | 4       | 0.0013  | 0.0018 | 0.0006 |
|               | 5       | 0.0032  | 0.0018 | 0.0005 |
|               | average | 0.00146 | 0.0018 | 0.0005 |

- SDR is based on NUIST, but main study on Fudan, why not do two simulations with the finest resolution for both stations?

Considering the complex underlying surface in urban Shanghai, we applied 4-domain-nested simulations to further study the Shanghai urban area, and the simulated results showed that there is no significant difference in the variations of amines. For the NUIST site which is located in a suburban area without complex underlying surface, 3-domain-nested simulations appear to be adequate.

- Model description must be improved, now the authors only say they follow Yu & Luo (2014), but this is the first time of implementing amine compounds in WRF-Chem, it needs to be explained in detail.

Absolutely necessary information:

- What is the particle uptake mechanism for amines?

- What are the oxidation coefficients? and which oxidants?

- other removal mechanisms? wet depositon for example?

More detailed model description and references have been added to Section 2.3.

- NMBs in Table 4 are not correct, it looks like that they are only bias of the total mean

 $\left(\frac{\overline{C_m}-\overline{C_0}}{\overline{C_0}}\right)$ . Correct way to calculate NMB is  $\frac{\sum_{i=1}^{N}(c_m-c_o)}{\sum_{i=1}^{N}c_o}$ , check Boylan & Russell (2006) for more information. As it is now, it can give a wrong impression of model ability to reproduce observations.

We calculated the NMBs according to the equation:

$$\frac{\sum_{i=1}^{n} (S_i - O_i)}{\sum_{i=1}^{n} O_i} \times 100\%$$

- It would be reasonable to focus on NUIST since the emission (SDR) factors are based on this station, so could you make run with the domain4 also for NUIST

Please see our reply to a similar comment earlier.

- Please add domain 3 for Fudan in Table 5 also, to facilitate comparing to NUIST site Added.

- Please analyse the discrepancy between model and observations more carefully, now the reasons for discrepancies are vague

The analysis has been improved and expanded by including the impact of uptake coefficients ( $\gamma$ =0.001, 0.01, 0.03) on the results.

- in addition to separating Fudan by agricultural/residential sector, add separation by land/sea also. This would allow evaluating non-pollution sector concentrations.

In the present study, the emission of amines over ocean is limited to ships and has already been included in the transport sector (Figs. 2-4e). The simulated concentrations of amines over ocean are generally quite lower.

- The sensitivity test is doubling/halving SDRs only. Can you use the uncertainty from observations to create uncertainty range in SDR and do sensitivity test with max/min range for that, could you do different particle uptake coefficients, this would be interesting. This way we could have an idea

**to which of the uncertainties are in most urgent need of new research.**

This is a good point, but the number of plumes for different sources is too limited to derive uncertainty range from observations. To look into impact of different particle uptake coefficients is a good suggestion. We have added two additional uptake coefficients ( $\gamma$ =0.03, 0.01), and relevant discussion has been given in Section 3.2 of the revised manuscript.

- Authors refer to short lifetime for amine many times without a reference or calculation of lifetime of amines, please add reference and/or calculation from your model

Two references have been added.

- Can you compare the particle size distributions with observations to evaluate the particle sink for amine?

Particle size distributions were not observed at the two sites during the periods when amines were measured.

**High resolution modeling of gaseous methylamines over a polluted region in China: Source-dependent emissions and implications to spatial variations**

Jingbo Mao1, Fangqun Yu1, 2\*, Yan Zhang1\*, Jingyu An3, Lin Wang1, Jun Zheng4, Lei Yao1, Gan

5 Luo2, Weichun Ma1, Qi Yu1, Cheng Huang3, Li Li3, and Limin Chen1

[revised manuscript text omitted]

5 SDR for domain 3.

I

|                   | Ammonia | C1-amine | C2-amines | C3-amines |   |
|-------------------|---------|----------|-----------|-----------|---|
| agriculture       | 785.20  | 460.73   | 444.94    | 97.28     | - |
| residential       | 103.09  | 62.19    | 389.47    | 16.34     |   |
| transportation    | 23.19   | 13.48    | 7.88      | 3.01      |   |
| other industry    | 7.47    | 6.15     | 5.08      | 1.08      |   |
| chemical industry | 0.65    | 9.32     | 1.73      | 0.08      |   |
| Total             | 919.61  | 551.88   | 849.11    | 117.78    |   |

Notes: the unit of ammonia: Gg (N) yr-1, the unit of C1-, C2-, and C3-amines: Mg (N) yr-1

Table 4. Statistical performance methylamines simulation at NUIST site (FR2012, SDR2012) in domain 3 and Fudan site (FR2015, SDR2015) in both domain 3 and domain 4 (values given in parentheses).

| Case             | Variable  | No.samples | Obs.ave | Sim.ave
(Domain4) | NMB (Domain4)   |
|------------------|-----------|------------|---------|----------------------|-----------------|
|                  | C1-amine  | 61         | 4.35    | 8.97                 | 106.72          |
| NUIST
FR2012  | C2-amines | 61         | 7.08    | 1.99                 | -71.50          |
|                  | C3-amines | 61         | 1.91    | 8.64                 | 359.02          |
|                  | C1-amine  | 61         | 4.35    | 6.39                 | 45.60           |
| NUIST
SDR2012 | C2-amines | 61         | 7.08    | 10.56                | 49.12           |
|                  | C3-amines | 61         | 1.91    | 1.12                 | -41.26          |
|                  | C1-amine  | 719        | 15.71   | 6.79 (9.26)          | -56.75 (-41.03) |
| Fudan
FR2015  | C2-amines | 719        | 40.20   | 1.56 (2.15)          | -96.13 (-94.67) |
|                  | C3-amines | 719        | 1.13    | 6.71 (9.24)          | 494.28 (718.61) |
| Fudan
SDR2015 | C1-amine  | 719        | 15.71   | 4.97 (6.61)          | -68.37 (-57.95) |
|                  | C2-amines | 719        | 40.20   | 16.33 (25.15)        | -59.37 (-37.43) |
|                  | C3-amines | 719        | 1.13    | 1.01 (1.37)          | -10.84 (21.34)  |

Notes: the unit of Obs.ave and Sim.ave: pptv, the unit of NMB: %

Table 5. Variations in normalized mean bias (NMBs) of methylamines simulations when amines emission rates are halved or doubled, at NUIST site (SDR2012, 0.5 SDR2012, 2 SDR2012) in domain 3 and Fudan site (SDR2015, 0.5 SDR2015, 2 SDR2015) in domain3 and domain 4 (values given in parentheses).

| Sensitivity Case | C1-amine       | C2-amines      | C3-amines      |
|------------------|----------------|----------------|----------------|
| SDR2012          | 45.60          | 49.12          | -41.26         |
| 0.5 SDR2012      | -27.96         | -25.73         | -74.31         |
| 2 SDR2012        | 193.13         | 199.01         | 88.97          |
| SDR2015          | -74.95(-51.23) | -58.07(-9.73)  | -17.13(69.62)  |
| 0.5 SDR2015      | -81.33(-75.99) | -69.82(-55.10) | -42.07(-28.51) |
| 2 SDR2015        | -24.23(-2.00)  | 21.19(80.34)   | 388.59(513.09) |

Table 6. Variations in normalized mean bias (NMBs) of methylamines simulations when aerosol uptake coefficients is 0.001, 0.01, and 0.03 at Fudan site in domain 4 during period from 25 July and 31 July 2015

| Uptake
coefficients(γ) | C1-amine | C2-amines | C3-amines |
|---------------------------|----------|-----------|-----------|
| 0.001                     | -51.23   | -9.73     | 69.62     |
| 0.01                      | -57.17   | -14.98    | 54.79     |
| 0.03                      | -64.33   | -23.02    | 31.84     |

Figure 1. Four nested domains in the present study. Domain 1 covers East-Asia and part of south-east Asia. Nested domain 2, 3, and 4 cover a large part of East- China, the Yangtze River Delta (including Nanjing, Shanghai), and Shanghai, respectively.

---

## Author Response (AR2)

We would like to thank both reviewers for their further comments which help to improve our manuscript. Our point-to-point replies (in blue) to the comments are given below (the original comments are copied here in black). The manuscript has been revised accordingly. All the changes to the manuscript have been highlighted using the Microsoft word "track-changes" tool in one version of the submitted revised manuscript.

*Anonymous Referee #2*

Suggestions for revision or reasons for rejection (will be published if the paper is accepted for final publication)

In your Reply, you indicate that small differences in ratios do not affect the conclusions. In a scientific article you need to represent all data correctly, so that the reader has the proper information to replicate and/or base his/her future research on.

As we emphasized in the manuscript, there exists large uncertainty in the model representation of amine emissions and there is no established approach to estimate amine emissions. We think the peak values in observations could represent the typical emission profiles well. Thus, we chose using the peak values to create the ratios of amines to ammonia as a reference to build the emission inventory in this study. We find the ratios are comparable with those reported in Zheng et al. (2015) and the differences are well within the uncertainty. We have clarified this in the text.

Some revisions are needed (page and line numbers refer to the authors response in the review system, not the public version):

1).Table 1 note in the caption that the numbers here are the peak values in Zhang et al. Fig. 6.

Revised as suggested.

2).Table 2 add information for sensitivity runs (gamma)

Done.

3).Tables 4 and 5 have different numbers for Fudan site, please correct the numbers.

The different numbers are expected as Table 4 shows the NMBs for the simulations from July 25 to 25 August 2015, while the numbers shown in Table 5 are calculated for the period from 25 July to 31 July 2015 when model simulated relatively well the wind fields.

4).Table 4 caption should indicate which run it is.

Revised.

5).P16L23 Change derived to actual method used, e.g. calculated using orthogonal distance regression. Please explain the rationale using only the peak values.

Please see our reply to a similar comment earlier, and relevant sentence has been added in Section 2.2 of the revised manuscript.

6).P17L22 Why OH only? Please add explanation.

Done.

7).P17L24 Besides gamma please explain how the uptake is calculated, gamma alone is not enough.

Revised.

8).P18L1 add information for the uptake coefficient for sensitivity runs as well (as in Table 2).

Added.

9).P18L15-19 Please add information for the sensitivity runs with different gamma.

Added.

10).P21L23-P22L3 Add discussion how the NMB changes when you change the gamma.

Added.

11).P21L28 night times -> night time (remove s)

Revised.

12).P21L29 Please refer to NMB or Fig to support the better agreement with obs.

Modified.

13).P22L2 This indicates that the uptake coefficient could be different amines, which is a result in itself. Note it here and add to conclusions.

Modified.

14).P22L8 'It is clear' is a bit vague; based on what? Please use rather something like: Figure shows.

Modified.

15).P22L14 Please indicate the actual NMB for FR instead of 'under -90%'.

Modified.

16).P25L25 Add a line explaining different uptake between amine types.

Modified.

*Anonymous Referee #3*

Suggestions for revision or reasons for rejection (will be published if the paper is accepted for final publication)

This paper describes modeling simulations of amine emissions in Nanjing-Shanghai region in China, and compares the simulations with measured amine concentrations. Measurements and simulations of amines and amine emissions are extremely challenging, although they are much needed. The authors want to see the relative differences in emissions of C1-C3 amines from different emissions sources. Considering recent publications that show exceedingly high concentrations of amines from this region, this work is also timely. I recommend publication with minor revisions.

We appreciate the positive comments confirming the importance of this study.

Only ammonia has been used in global models, even though representation of amine concentrations and emissions is still very poor. Because often amines are emitted together ammonia, it would make sense to attempt to derive the ratios of amines vs. ammonia emissions. But I would consider that a separate treatment of amines and ammonia emissions will be more useful, and models then can use ammonia data to constrain amines. Because ammonia itself it so little understood, using ammonia as a reference does not improve much modeling predictions and this only carries high uncertainties in modeling.

This is a good point, but the measurements for amines are too limited in China to derive meaningful amines emissions. As pointed in the Introduction, fixed amines to ammonia ratios were used to estimate amines emissions in global models. The present study is the first attempt (to our knowledge) to use direct measurements to constrain amine emissions from different sources. The SDR approach, as we show here, improves the skill of the model in simulating concentrations of amines in polluted regions. We hope more field observations as well as more accurate source apportionment of amines will be carried out in the future to constrain the amines emissions and model study.

Please explain how you derived emissions from amine concentrations.

We derived methylamines emissions based on amines to ammonia ratios and ammonia emissions from different sources. The temporal and spatial distributions of C1-, C2-, and C3-amines follow those of ammonia for different sources.

It seems general that C3-amines are more abundant, at least in several U.S. locations and at indoor, so I am surprised by lower emissions of C3-amines from this model in this particular region.

The measurements at both NUIST and Fudan site in China showed the concentrations of C2-amines are the highest. Zheng et al. (2015) pointed out that TMA maybe overestimated caused by the impact of acetamide with the same nominal mass TMA in previous study.

The paper could have been written more concisely, but I believe this is up to each individual's style.

Thanks for the comment. The manuscript has been revised by my multiple co-authors and we hope that the results presented will be useful to the community.

[revised manuscript text omitted]